# The SARS-CoV-2 Delta variant induces an antibody response largely focused on class 1 and 2 antibody epitopes

**Allison J. Greaney[1,2]\*, Rachel T. Eguia[1], Tyler N. Starr[1,3], Khadija Khan[4,5], Nicholas Franko[6], Jennifer K. Logue[6], Sandra M. Lord[7], Cate Speake[7], Helen Y. Chu[6], Alex Sigal[4,5,8,9], Jesse D. Bloom[1,3]\***

**1** Basic Sciences Division and Computational Biology Program, Fred Hutchinson Cancer Research Center, Seattle, Washington, United States of America, **2** Department of Genome Sciences & Medical Scientist Training Program, University of Washington, Seattle, Washington, United States of America, **3** Howard Hughes Medical Institute, Chevy Chase, Maryland, United States of America, **4** Africa Health Research Institute, Durban, South Africa, **5** School of Laboratory Medicine and Medical Sciences, University of KwaZulu–Natal, Durban, South Africa, **6** Division of Allergy and Infectious Diseases, University of Washington, Seattle, Washington, United States of America, **7** Center for Interventional Immunology, Benaroya Research Institute at Virginia Mason, Seattle, Washington, United States of America, **8** Centre for the AIDS Programme of Research in South Africa, Durban, South Africa, **9** Max Planck Institute for Infection Biology, Berlin, Germany

\* agreaney@fredhutch.org (AJG); jbloom@fredhutch.org (JDB)

**Data Availability Statement:** All relevant data are within the manuscript and its Supporting Information files. The complete code for the full computational data analysis pipeline of the

## Abstract

Exposure histories to SARS-CoV-2 variants and vaccinations will shape the specificity of antibody responses. To understand the specificity of Delta-elicited antibody immunity, we characterize the polyclonal antibody response elicited by primary or mRNA vaccine-breakthrough Delta infections. Both types of infection elicit a neutralizing antibody response focused heavily on the receptor-binding domain (RBD). We use deep mutational scanning to show that mutations to the RBD's class 1 and class 2 epitopes, including sites 417, 478, and 484–486 often reduce binding of these Delta-elicited antibodies. The anti-Delta antibody response is more similar to that elicited by early 2020 viruses than the Beta variant, with mutations to the class 1 and 2, but not class 3 epitopes, having the largest effects on polyclonal antibody binding. In addition, mutations to the class 1 epitope (e.g., K417N) tend to have larger effects on antibody binding and neutralization in the Delta spike than in the D614G spike, both for vaccine- and Delta-infection-elicited antibodies. These results help elucidate how the antigenic impacts of SARS-CoV-2 mutations depend on exposure history.

## Author summary

SARS-CoV-2 has been circulating among humans for over two years, and immune selection has become a major driver of its evolution. In addition to partially evading the immune response elicited by vaccination or infection with earlier variants, SARS-CoV-2 "variants of concern" can also elicit distinct antibody responses. Here, we use the high-throughput techniques of deep mutational scanning and yeast display to characterize the

mapping experiments is available at https://github.com/jbloomlab/SARS-CoV-2-RBD_Delta. • The neutralization titers of vaccine- and infection-elicited sera against the tested RBD point mutants are in S1 Data and at https://github.com/jbloomlab/SARS-CoV-2-RBD_Delta/blob/main/experimental_data/results/neut_titers/combined_neut_titers.csv. • All raw sequencing data are available on the NCBI Short Read Archive at BioProject PRJNA770094, BioSample SAMN25941479 (PacBio CCS reads), SAMN25944367 (Illumina reads for ACE2 binding and expression experiments) and SAMN26317640 (antibody-escape experiments). • The ACE2 binding and expression scores for mutations in the Delta background are in S2 Data and at https://github.com/jbloomlab/SARS-CoV-2-RBD_Delta/blob/main/results/final_variant_scores/final_variant_scores.csv. • The escape fractions measured for each mutation are in S3 Data and also at https://github.com/jbloomlab/SARS-CoV-2-RBD_Delta/blob/main/results/supp_data/aggregate_raw_data.csv. • Interactive visualizations of the mutations that reduce binding of polyclonal antibodies to the Delta RBD are here: https://jbloomlab.github.io/SARS-CoV-2-RBD_Delta/.

**Funding:** This project has been funded in part with federal funds from the NIAID/NIH (https://www.niaid.nih.gov/) under contract numbers HHSN272201400006C and 75N93021C00015 to J.D.B. This work was supported by grants from the NIAID / NIH (R01AI141707 to J.D.B., T32AI083203 to A.J.G., and R01 AI138546 to A.S.), and the Gates Foundation (https://www.gatesfoundation.org) (INV-004949 to J.D.B, INV-016575 to H.Y.C., and INV-018944 to A.S.). The Scientific Computing Infrastructure at Fred Hutch is funded by ORIP (https://orip.nih.gov/) grant S10OD028685. T.N.S. is a Howard Hughes Medical Institute Fellow of the Damon Runyon Cancer Research Foundation (https://www.damonrunyon.org/) (DRG-2381-19). J.D.B. is an Investigator of the Howard Hughes Medical Institute (https://www.hhmi.org/). H.Y.C. is also funded by an Emergent Ventures Award (https://www.mercatus.org/emergent-ventures). COVID studies at BRI (C.S., S.M.L.) were supported by the Benaroya Family Foundation, the Leonard and Norma Klorfine Foundation, Glenn and Mary Lynn Mounger, and Monolithic Power Systems. The funders had no role in study design, data collection and analysis, decision to publish, or preparation of the manuscript.

**Competing interests:** I have read the journal's policy and the authors of this manuscript have the following competing interests: J.D.B. consults for Moderna and Flagship Labs 77 on topics related to

antibody response elicited by primary or vaccine-breakthrough infections with the Delta variant. We compare these responses to those elicited by mRNA vaccination, infection with early 2020 viruses, or infection the Beta variant. We find that the early 2020 and Delta-elicited antibody responses often target similar regions of the spike receptor-binding domain, with some subtle but notable differences. The antibody response to Delta and early 2020 strains are more similar to each other than to the Beta variant. As SARS-CoV-2 continues to evolve, individuals' increasingly diverse exposure histories to vaccines and variants will influence their susceptibilities to future viral mutations.

## Introduction

New SARS-CoV-2 variants can not only circumvent preexisting immunity [1–11], but also elicit an antibody response that is different from that elicited by prior variants [4,12–17]. The SARS-CoV-2 Delta variant (B.1.617.2) rose to high global frequency in mid-2021 and was the dominant circulating variant [1,6,18] before being displaced by the Omicron variant (B.1.529) in late 2021 [18,19]. Delta caused a large wave of SARS-CoV-2 infections globally [18], including many breakthrough infections in individuals who had previously received a vaccine based on a strain of SARS-CoV-2 that circulated early in the pandemic [20,21].

Many people have thus been exposed to Delta as either a primary or breakthrough infection. It is important to understand the specificity of the antibody response elicited by these infections. Other studies have used antigenic cartography to dissect the antigenic relationships among SARS-CoV-2 variants by performing neutralization assays with serum from individuals who were vaccinated or had presumed primary exposures to different variants [15,17]. These studies provide important information about how well antibodies elicited by one variant cross-react with other variants. However, these studies primarily assay known variants with limited numbers of mutations and therefore do not identify which future mutations may further erode antibody immunity.

To identify which mutations have the potential to reduce binding of polyclonal antibodies elicited by primary or breakthrough Delta infection, we used deep mutational scanning [22] to measure the effect of every mutation in the Delta RBD on antibody binding. We compare the specificity of Delta-elicited antibodies to those elicited by earlier SARS-CoV-2 variants, including the early 2020 (i.e., Wuhan-Hu-1 and D614G) and Beta variants [12,23,24]. We find that all exposure histories, including Delta breakthrough infections, induce a neutralizing antibody response that primarily targets the RBD, and that within the RBD, Delta elicits antibodies that primarily target the class 1 and 2 epitopes.

## Results

### The Delta variant contains multiple mutations in spike and dominated global circulation in mid-2021

The Delta variant (B.1.617.2) rose to high frequency among globally circulating SARS-CoV-2 viruses in 2021 [1,6,18]. Compared to the Wuhan-Hu-1 prototypical early 2020 virus, Delta has multiple mutations in the spike protein: T19R, Δ157–158, L452R, T478K, D614G (which fixed in circulating SARS-CoV-2 isolates in mid-2020 [25]), P681R, and D950N (**Fig 1A**) [6]. Two of these mutations, L452R and T478K, are in the spike RBD.

The SARS-CoV-2 Beta variant (B.1.351) spike, which we also examine here, also has a number of mutations, including K417N, E484K, and N501Y mutations in the RBD (**Fig 1B**) [26].

viral evolution, and has the potential to receive a share of IP revenue as an inventor on a Fred Hutch optioned technology/patent (application WO2020006494) related to deep mutational scanning of viral proteins. A.J.G, T.N.S., and J.D.B have the potential to receive a share of IP revenue as an inventor on a Fred Hutch-optioned technology related to deep mutational scanning of the receptor-binding domain of SARS-CoV-2 Spike protein. H.Y.C. is a consultant for Merck, Pfizer, Ellume, and the Bill and Melinda Gates Foundation and has received support from Cepheid and Sanofi-Pasteur. The other authors declare no competing interests.

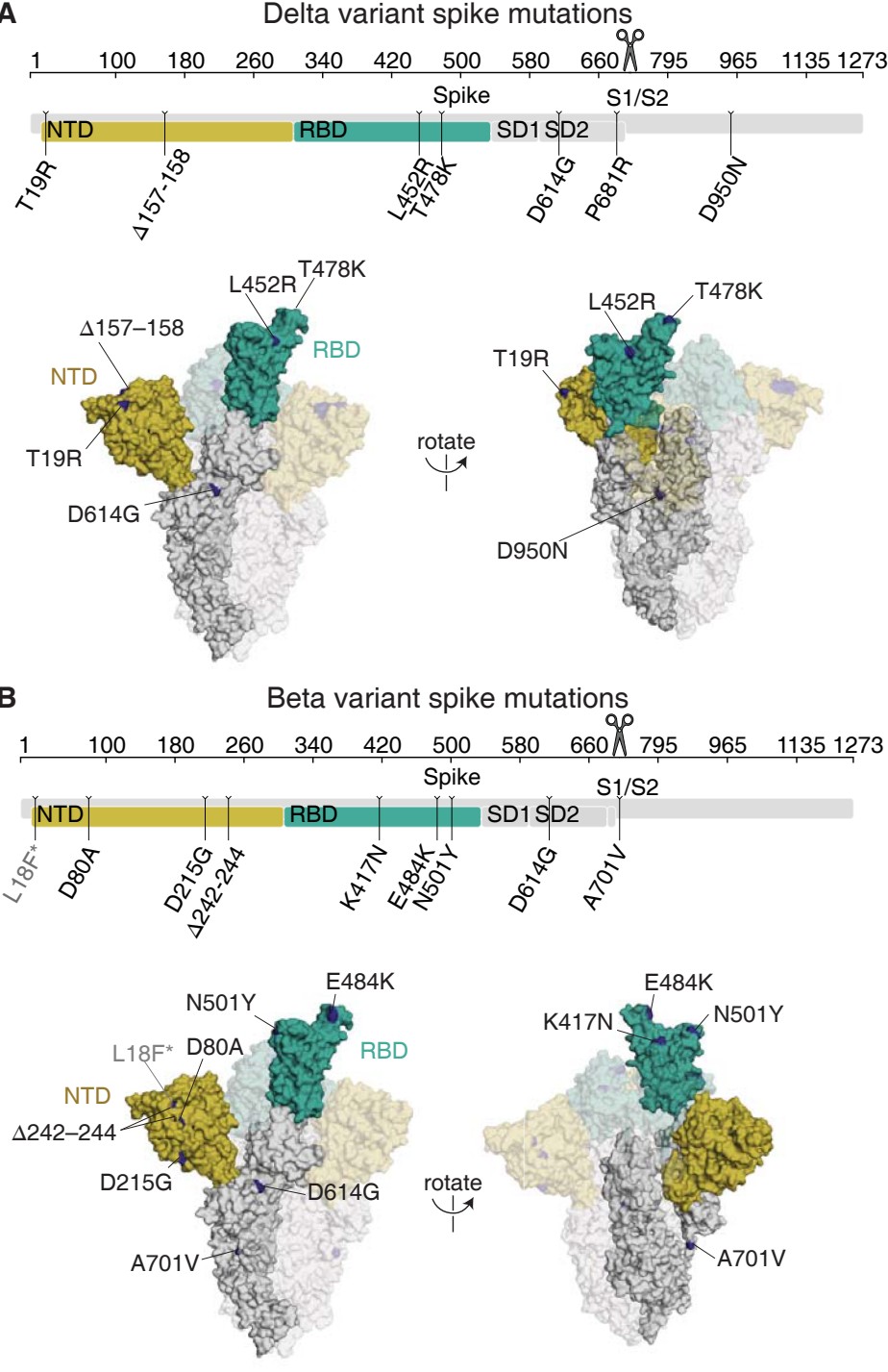

**Fig 1. Delta and Beta spikes contain mutations in multiple domains. (A,B)** Mutations in the Delta (A) or Beta (B) spikes relative to Wuhan-Hu-1 [6,26]. Sites where mutations occur in the spike ectodomain are highlighted in dark blue on the Wuhan-Hu-1 one-RBD open spike trimer (PDB 6ZGG) [63]. The surface of one spike monomer is shown; the other two protomers are transparent. Visualization of linear spike sequence modified from https://covdb.stanford.edu/sierra/sars2/by-patterns/. Scissors icon by Mario Verduzco from NounProject.com.

Some of the RBD and NTD mutations in Delta and Beta are in antigenic sites [24,27,28] and both variants have reduced sensitivity to neutralization by early 2020 viral infection- or vaccination-elicited antibodies—although the reduction in neutralization is larger for Beta [1,4,13].

### Description of cohorts

To study the antibody response elicited by a primary Delta infection, we obtained plasma samples from individuals convalescent of a primary Delta infection without a history of prior SARS-CoV-2 infection or vaccination. The samples were collected in South Africa from August–September 2021 when the Delta variant was predominant, approximately 30 days post-symptom onset (mean 33.4, range 24–37) (**S1 Table**). We also obtained plasma samples from individuals who had completed a two-dose series of an mRNA vaccine and were subsequently infected with the Delta variant (Delta breakthrough infection). These samples were collected in Washington State, USA, from July–September, 2021 (**S1 Table**). We compared these samples to previously published studies of samples collected approximately 30 days post-symptom onset of infection with an early 2020 SARS-CoV-2 variant [23,24] or the Beta (B.1.351) variant [12] (**S1 Table**). We also compare some neutralization assays to those previously measured in individuals who completed a two-dose series of mRNA-1273 vaccination [29].

### Most neutralizing activity elicited by primary SARS-CoV-2 infection or vaccination is directed towards the RBD

Primary SARS-CoV-2 infection or mRNA vaccination generally elicits a neutralizing antibody response that primarily targets the spike RBD [12,23,30,31]. To confirm this finding for antibodies elicited by the Delta variant, we depleted primary Delta infection convalescent plasmas of antibodies that bind to the Delta RBD, or performed a mock depletion, and measured the residual neutralizing activity against lentiviral particles pseudotyped with the Delta spike. For all 8 tested plasma samples, removal of Delta RBD-binding antibodies reduced the neutralization potency by 48-fold or more (**Figs 2A and S1 and S2 and S1 Data**).

Taken in combination with our previous studies, these results show that early 2020 [23], Beta [12], and Delta variant infections all elicit a primarily RBD-focused neutralizing antibody response, although some neutralizing antibodies also target non-RBD epitopes (**Fig 2A**). mRNA vaccination, however, elicits a neutralizing antibody response that is hyper-focused on the RBD, even more so than infection (**Fig 2A**, right panels) [29]. This may be due to differential presentation of antigens and conformations of the spike protein in vaccination versus infection.

### RBD-binding antibodies also dominate the neutralizing antibody response elicited by Delta breakthrough infections

To understand the effect of a Delta breakthrough infection after mRNA vaccination, serum samples from individuals who had completed a two-dose series of BNT162b2 vaccination (hereafter referred to as 2x BNT162b2 vaccination) or who had a Delta breakthrough infection were depleted of D614G or Delta RBD-binding antibodies and assessed for neutralizing activity against the homologous spike. The Delta-specific neutralizing antibody response elicited by 2x BNT162b2 vaccination is entirely focused on the RBD (**Fig 2B**, third panel from left), consistent with results reported by others [32]. As expected [33], Delta breakthrough infection resulted in higher neutralizing titers against both D614G (by ~4.8-fold) and Delta (by ~3.8-fold) spikes than 2x BNT162b2 alone. The neutralizing antibody activities elicited by a

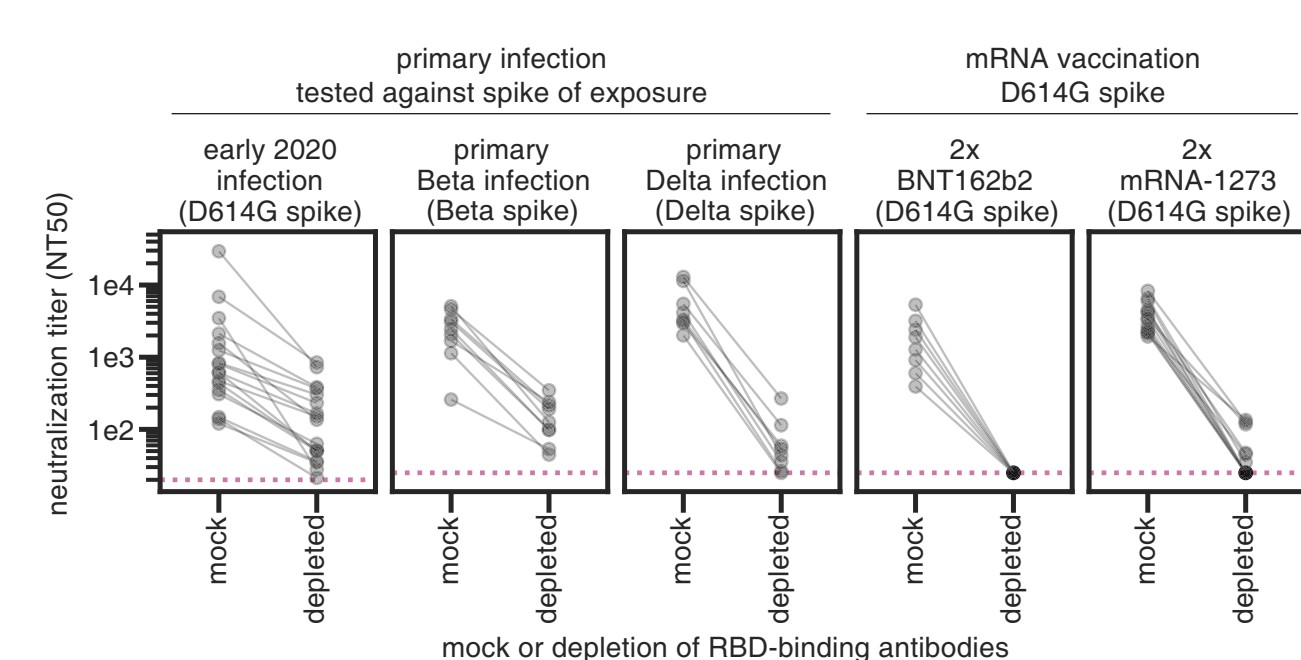

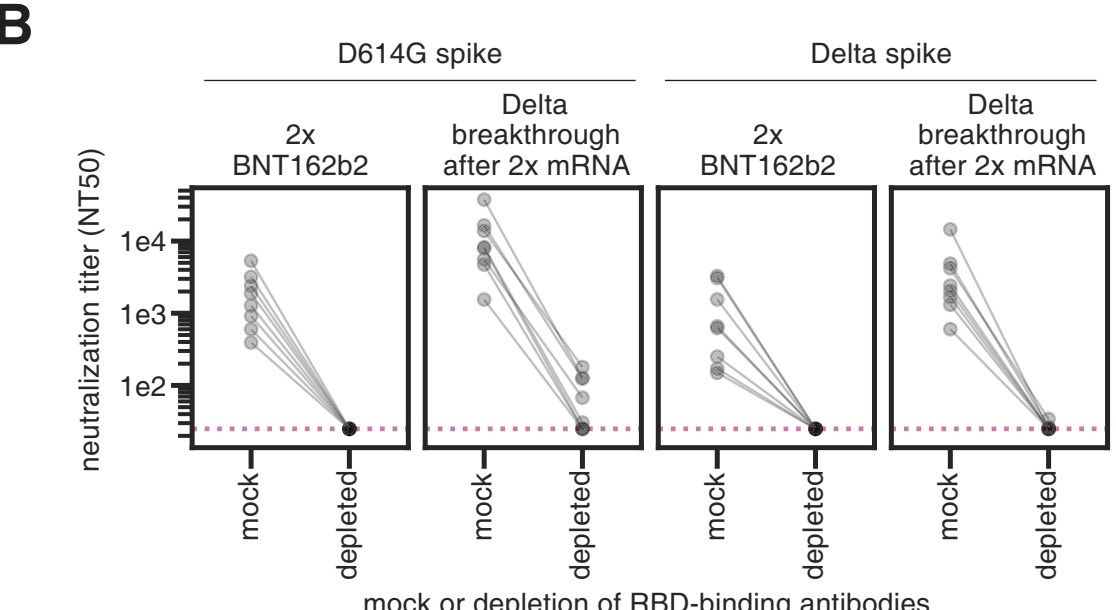

**Fig 2. SARS-CoV-2 infection or vaccination elicits a neutralizing antibody response highly focused on the RBD. (A)** Plasma neutralization (neutralization titer 50%, NT50) against lentiviral particles pseudotyped with the homologous spike of exposure (in parentheses) following either mock depletion or depletion of RBD-binding antibodies for individuals who had primary infections with early 2020, Beta, or Delta viruses (and no prior SARS-CoV-2 infections or vaccinations), or individuals who completed the two-dose series of the mRNA vaccines BNT162b2 or mRNA-1273 (labeled 2x BTN162b2 or 2x mRNA-1273). **(B)** Plasma neutralization following either mock depletion or depletion of RBD-binding antibodies for individuals who were vaccinated 2x with BNT162b2 or who had Delta breakthrough infections after 2x mRNA vaccination. Plasma samples were depleted of D614G RBD-binding antibodies and tested for neutralization against D614G spike (left), or depleted of Delta RBD-binding antibodies and tested for neutralization against Delta spike (right). In all cases, the RBD and spike from the same viral variant were used in both the depletions and neutralization assays. Note that the same 2x BNT162b2 data are shown in the right panel of (A) and in the left panel of (B) to facilitate visual comparison. The neutralizing titers for the early 2020 plasmas were first published in [23], the Beta plasmas in [12], and the 2x mRNA-1273 plasmas in [29]. The assays for the primary Delta infection, Delta breakthrough infection, and 2x BNT162b2, with both D614G and Delta RBD and spike were newly performed in this study. ELISAs that confirm depletion of RBD-binding antibodies performed in this study are in **S1 Fig**. The results new to this study are plotted by individual serum or plasma in **S2 Fig**. Full neutralization curves are shown in **S7 Fig**.

Delta breakthrough infection against the D614G and Delta spikes are also primarily focused on the RBD (**Fig 2B**, second and fourth panels from left, respectively). While Delta breakthrough infection does boost modest non-RBD neutralizing activity against the D614G spike (**Fig 2B**, second panel from left), the Delta spike-reactive neutralizing activity is entirely directed towards the RBD (**Fig 2B**, fourth panel from left).

Together, these results show that the majority of Delta-specific neutralizing activity elicited by Delta infection, mRNA vaccination, or Delta breakthrough infection is directed towards the RBD. A caveat is that all neutralization assays were performed on cells that overexpress ACE2, which emphasize the effect of RBD-binding antibodies versus those targeting other regions of spike [3,34,35].

## Deep mutational scanning to comprehensively measure mutations that reduce antibody binding to the Delta RBD

To understand which RBD mutations may erode Delta-elicited antibody binding, we used a previously described yeast-display deep mutational scanning method to comprehensively identify mutations to the Delta RBD that reduce antibody binding [22,23]. We generated duplicate mutant libraries comprising all possible mutations in the Delta RBD and measured the effects of mutations on RBD expression and ACE2 binding (**S3 Fig** and **S2 Data**)[36]. We used both computational and experimental filters to remove mutations that could present as spurious antibody-escape mutations due to their highly deleterious effects on RBD folding or expression (**S3 Fig**).

To identify mutations that reduce binding of polyclonal antibodies to the Delta RBD, we incubated the yeast-displayed mutant libraries with each plasma and used FACS to enrich for mutants with reduced antibody binding, detected with an anti-human IgG+IgA+IgM secondary antibody (**S4 Fig**). We deep sequenced the pre- and post-selection populations to quantify each mutation's "escape fraction". These escape fractions range from 0 (no cells with the mutation in the plasma-escape bin) to 1 (all cells with the mutation in the plasma-escape bin) (**S3 Data**). We calculate the site-level antibody escape as the sum of the escape fractions for each mutation at a site (with a possible range from 0 to 19). The antibody-escape experiments were performed with two biologically independent replicate libraries. The replicate site- and mutation-level escape fractions were well-correlated (**S5 Fig**), and throughout we report the average across the two replicates (**S3 Data**). We represent the escape maps as logo plots, where the height of each letter is proportional to its escape fraction, or line plots showing the site-level escape (**Figs 3** and **S3A**).

## Primary Delta infection elicits a binding antibody response that is most affected by mutations to the class 1 and 2 epitopes

We find that a primary Delta variant infection elicits a polyclonal antibody response most focused on the class 1 and 2 epitopes in the Barnes classification scheme [37] (**Fig 3**). Mutations to sites 478 and 484–486 have some of the largest effects on antibody binding. Mutations to site 472 also impact binding in 6 out of 8 cases. For one plasma, mutations to the class 4 epitope and site P384 also reduce antibody binding.

Relative to Wuhan-Hu-1, the Delta RBD contains mutations in or proximal to the class 1 and 3 epitopes (T478K and L452R, respectively) (**Fig 3A**). While the L452R and T478K mutations affect binding and neutralization of some antibodies elicited by early 2020 viruses [1,2,29,38–41], only mutations to site K478, and not R452, sometimes strongly affect antibody binding of primary Delta infection-elicited plasmas (**Fig 3**). This is a relative reversal of what is

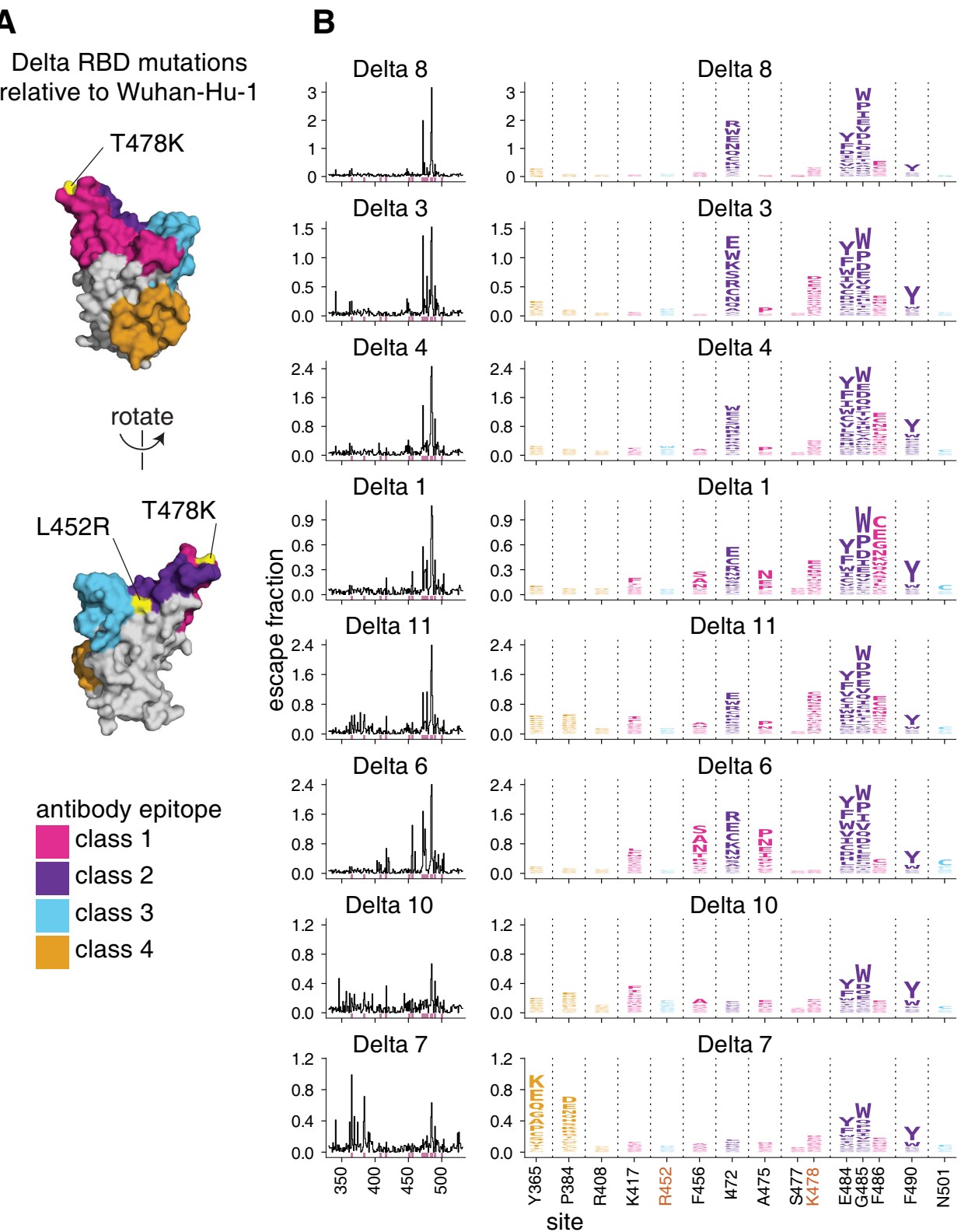

**Fig 3. Complete antibody-escape maps for plasmas from individuals convalescent of primary Delta infections.** (A) The surface of the Wuhan-Hu-1 RBD is colored according to antibody epitope per the Barnes classification scheme [37], with the L452R and T478K mutations in the Delta variant highlighted in yellow (and labeled with red text in panel B). We define L452R to be in the class 3 epitope, and T478K in the class 1 epitope. (B)

Complete maps of mutations that reduce binding of plasma antibodies to the Delta RBD from individuals convalescent of primary Delta infection. Sites of strong antibody escape (see Methods) for any of the 8 plasmas are highlighted with pink in the line plots at left and shown in the logo plots at right. Sites 452, 477, and 501 are included despite not being sites of strong escape due to their high frequency in circulating viral isolates. Site 408 is included to facilitate comparison to Delta breakthrough infection-elicited plasmas (Fig 5). Interactive versions of logo plots and structural visualizations are at https://jbloomlab.github.io/SARS-CoV-2-RBD_Delta/. Correlations between independent library replicates are in S5 Fig, and all escape scores are in S3 Data and online at https://github.com/jbloomlab/SARS-CoV-2-RBD_Delta/blob/main/results/supp_data/aggregate_raw_data.csv.

observed for early 2020-elicited antibodies, where mutations to site L452 have a larger effect than mutations to site T478 [23,24].

## Early 2020 viruses, Beta, and Delta elicit subtly different RBD antibody binding responses reflective of amino-acid differences in their RBDs

To understand how primary infection with different SARS-CoV-2 variants affects the specificity of the antibody response, we compared the above results for primary Delta infections to those previously measured for early 2020 or Beta variant infections [12,23,24] (Fig 4A and 4B). Infection with all three viruses elicits an antibody response that is affected by mutations to site 484 and the class 2 epitope (Figs 3 and 4A and 4B). Unlike the early 2020 and Beta variants, however, mutations to the class 3 epitope spanning residues 443–452 have little effect on binding by Delta-elicited antibodies. More similar to the early 2020 antibody response, Delta elicits a response affected by class 1 mutations, which had less effect on the Beta-elicited antibodies (Fig 4A and 4B).

To intuitively compare the binding-escape maps for the early 2020, Beta, and Delta variant cohorts, we used multidimensional scaling to project the complex antibody-escape maps into two dimensions (Fig 4C). We also included 22 previously characterized monoclonal antibodies representative of the 4 classes [22,24,42–44]. The multidimensional scaling projection (Fig 4C) shows that the early 2020 plasmas cluster between multiple antibody classes, but are somewhat biased towards class 2 antibodies, the Beta plasmas cluster between the class 2 and 3 antibodies, and the Delta plasmas cluster between the class 1 and 2 antibodies (Fig 4C). Overall, the binding-escape maps for Delta infections more closely resemble those for early 2020 infections than Beta infections. These results are consistent with serological evidence and antigenic cartography showing that the Beta and Delta variants are more distinct from one another than either is relative to the early 2020 variant [15–17,45,46].

## Delta breakthrough infection after two-dose mRNA vaccination elicits an antibody response that shares characteristics of both early 2020- and Delta primary infection responses

To understand how the antibody response to the Delta variant is affected by pre-existing immunity from vaccination, we repeated the antibody-escape mapping experiments with plasmas from individuals who were infected with Delta several months after 2x mRNA vaccination. For these experiments, we mapped antibody-binding escape mutations using the Delta RBD. These breakthrough plasmas, like the primary Delta infection plasmas, were focused on the class 1 and 2 epitopes, with mutations to sites 417 and 484–486 having some of the largest effects on antibody binding (Figs 5 and S6). Mutations to site K417 have larger effects on antibody binding for the Delta breakthrough plasmas than for the primary Delta infection plasmas, but mutations to site K478 have relatively smaller effects, reflective of Delta breakthrough antibody immunity being somewhat intermediate between early 2020 and primary Delta infection-elicited immunity (Figs 5 and S6). The Delta infection-elicited antibody response is even more focused on the class 1 and 2 epitopes than previously published antibody-escape maps of

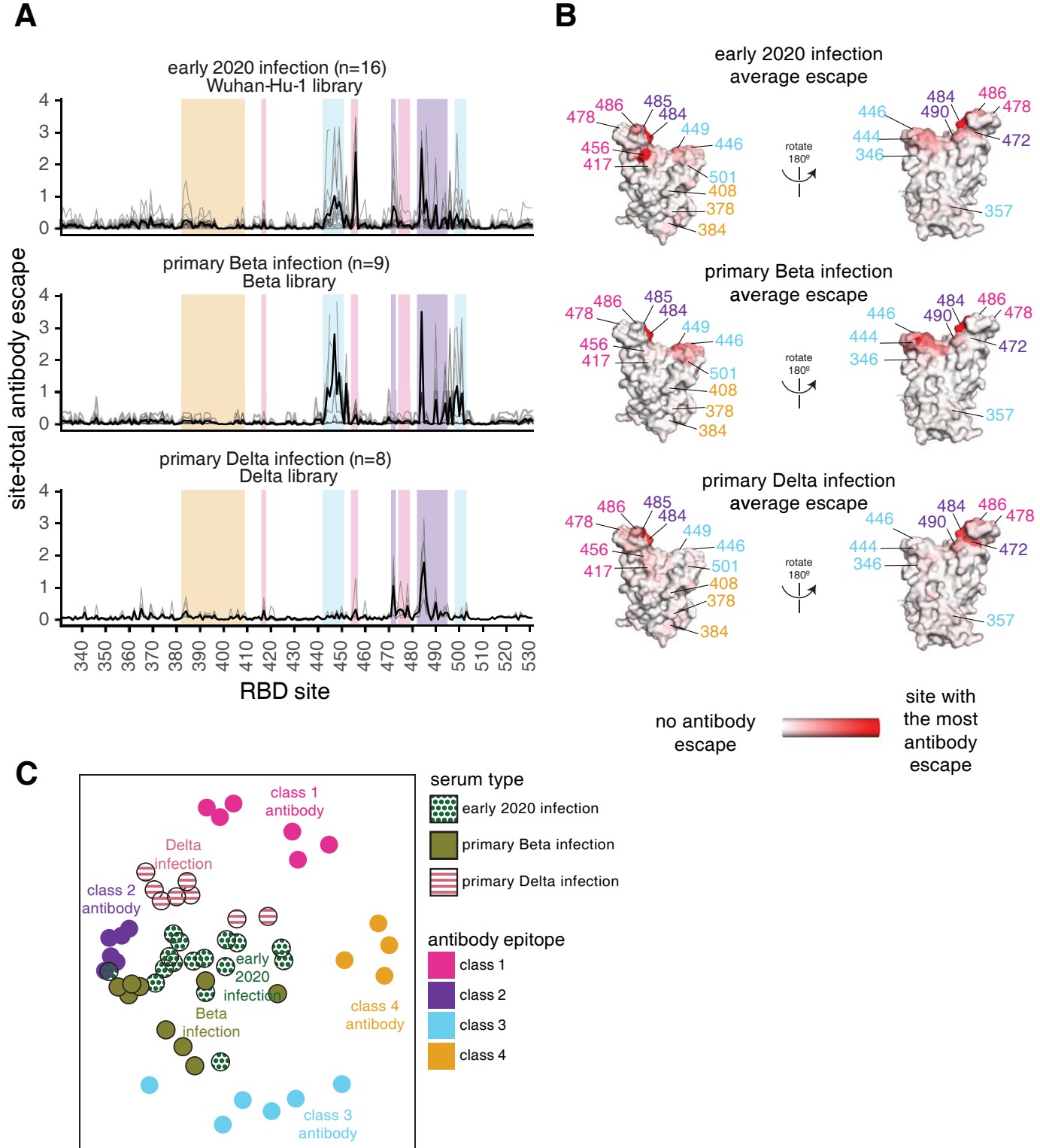

**Fig 4. Antibody-escape maps for individuals infected with early 2020 viruses, Beta, or Delta. (A)** Site-total antibody escape averaged across each group of plasmas for early 2020, Beta, and Delta convalescent plasmas, with key epitope regions highlighted (key in panel C). **(B)** Site-total antibody escape averaged across each group mapped to the Wuhan-Hu-1 RBD surface (PDB 6M0J, [64]), with red indicating the site with the most antibody escape and white indicating sites with no escape. Key antibody-escape sites are labeled, with site labels colored according to epitope. **(C)** Multidimensional scaling projection with the distance between points representing the dissimilarity between antibody-escape maps for individual early 2020, primary Beta, and primary Delta infection convalescent plasmas, with monoclonal antibody escape maps of the 4 classes used as anchors. In all panels, the convalescent plasmas were mapped against RBD mutant libraries corresponding to the viral variant of exposure. The antibody-escape maps for early 2020 infections were first reported in [23,24], the Beta infections in [12], and the monoclonal antibodies in [22,24,42–44].

The antibody-escape maps for primary Delta infections are new to this study. The complete antibody-escape scores are in **S3 Data** and online at https://github.com/jbloomlab/SARS-CoV-2-RBD_Delta/blob/main/results/supp_data/aggregate_raw_data.csv.

mRNA-1273 vaccine-elicited sera, which were relatively broad but skewed towards class 1 and 2 [29] (**Figs 5 and S6**).

## K417N has a larger effect on neutralization in the Delta than in the D614G spike

To determine whether the effects of mutations in the class 1 and 2 epitopes on antibody binding correspond to reductions in neutralization, we tested plasma neutralization against the canonical class 1 and 2 antibody-escape mutations K417N and E484K [24] in the Delta RBD background. We compared these results to the neutralization of D614G (early 2020) and Delta spike-pseudotyped lentiviral particles. We also included plasmas depleted of Delta RBD-binding antibodies to estimate the lower limit of neutralization for any possible RBD mutations (**Figs 6 and S7**).

As expected, the 2x BNT162b2 and Delta breakthrough plasmas most potently neutralized the D614G spike, and primary Delta infection plasmas most potently neutralized the Delta spike (**Figs 6 and S7**). The Delta + K417N mutation resulted in a ~3-fold reduction in neutralization for 2x BNT162b2 and Delta breakthrough plasmas when compared to the Delta spike. The primary Delta infection plasmas also had a ~2-fold reduced neutralization potency against Delta + K417N compared to Delta spike. This is in contrast to the lack of effect of the K417N mutation in the D614G spike background for early 2020 infection- or vaccine-elicited plasmas (**Fig 6B**). These results are consistent with other reports that K417N alone has little effect in the D614G background [3,7,12,29], although its effect can be unmasked when it is found in conjunction with mutations in more immunodominant sites such as E484K or L452R [1,15,47].

In the D614G background, the E484K mutation has a large effect on antibody neutralization for early 2020 and mRNA vaccination-elicited plasmas. In some cases, E484K alone can reduce neutralization to almost the same degree as removal of all RBD-binding antibodies (**Fig 6A**) [12,29]. Delta + E484K has an ~8-fold effect on neutralization for primary Delta-infection elicited antibodies, but only a ~3-fold effect for mRNA vaccination- or Delta breakthrough infection-elicited antibodies, comparable to the effect of Delta + K417N mutation. The L452R mutation likely disrupts binding of some 484-binding antibodies [42,48], probably explaining why the E484K mutation has a relatively smaller effect in the L452R-containing Delta spike. This interpretation is supported by findings from other groups that the L452R and E484Q mutations in the Kappa variant have a less-than-additive effect on neutralization [1,15,49].

Notably, sublineages of the Delta variant, AY.1 and AY.2, colloquially referred to as "Delta +" lineages, contain the K417N mutation [50]. Thus, mutations that further erode early 2020 infection and vaccine-elicited immunity (i.e., the K417N mutation) and mutations that erode Delta-elicited immunity (i.e., those to site 484) may have been under continued antigenic selection when the Delta variant was dominant.

## Discussion

In this study, we used deep mutational scanning and serological assays to study the antibody binding and neutralizing response elicited by Delta variant infection. The majority of neutralizing activity elicited by SARS-CoV-2 vaccination or infection targets the RBD [12,23,29]. We show that this is also true for primary and breakthrough Delta infection-elicited antibodies,

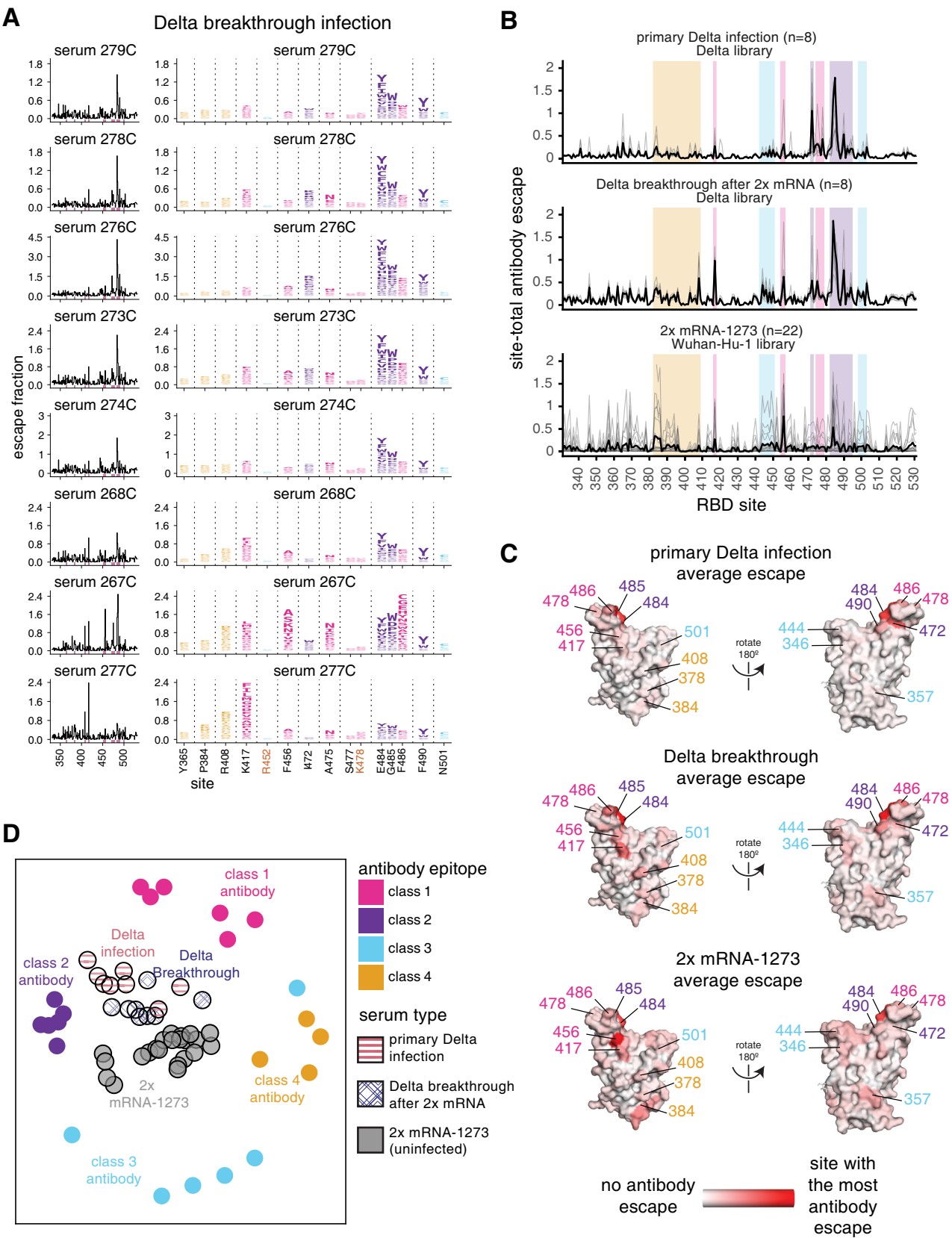

**Fig 5. Delta breakthrough infection after 2x mRNA vaccination elicits an antibody response focused on the class 1 and 2 epitopes. (A)** Complete maps of mutations that reduce binding of plasma antibodies to the Delta RBD from individuals convalescent of Delta breakthrough infection. The same sites as in **Fig 3B** are shown in the logo plots and highlighted in pink in the line plots. Logo plots for both cohorts are shown side-by-side in **S6 Fig. (B)** Site-total antibody escape averaged across each group of plasmas for primary Delta convalescent plasmas, Delta-breakthrough convalescent plasmas, and previously published mRNA-1273-vaccinated sera [29], with key epitope regions highlighted (key in panel D). **(C)** Site-total antibody escape averaged across each group mapped to the RBD surface, with red indicating the site with the most antibody escape and white indicating sites with no escape. The Delta primary infection and Delta breakthrough plasmas were mapped against the Delta RBD libraries, and the mRNA-1273 sera were mapped against the Wuhan-Hu-1 RBD libraries. Site labels are colored by epitope. **(D)** Multidimensional scaling projection with the distance between points representing the dissimilarity between antibody-escape maps, comparing primary Delta infections, Delta breakthrough after 2x mRNA vaccination convalescent plasmas, and mRNA-1273-vaccinated sera with monoclonal antibody escape maps of the 4 classes used as anchors, first reported in [22,24,42–44]. The antibody-escape maps for mRNA-1273 vaccine-elicited sera were first reported in [29], and primary Delta and Delta breakthrough infections are new to this study. The primary Delta infection escape maps are replicated from **Fig 4**. Interactive versions of logo plots and structural visualizations are at https://jbloomlab.github.io/SARS-CoV-2-RBD_Delta/. The complete antibody-escape scores are in **S3 Data** and online at https://github.com/jbloomlab/SARS-CoV-2-RBD_Delta/blob/main/results/supp_data/aggregate_raw_data.csv.

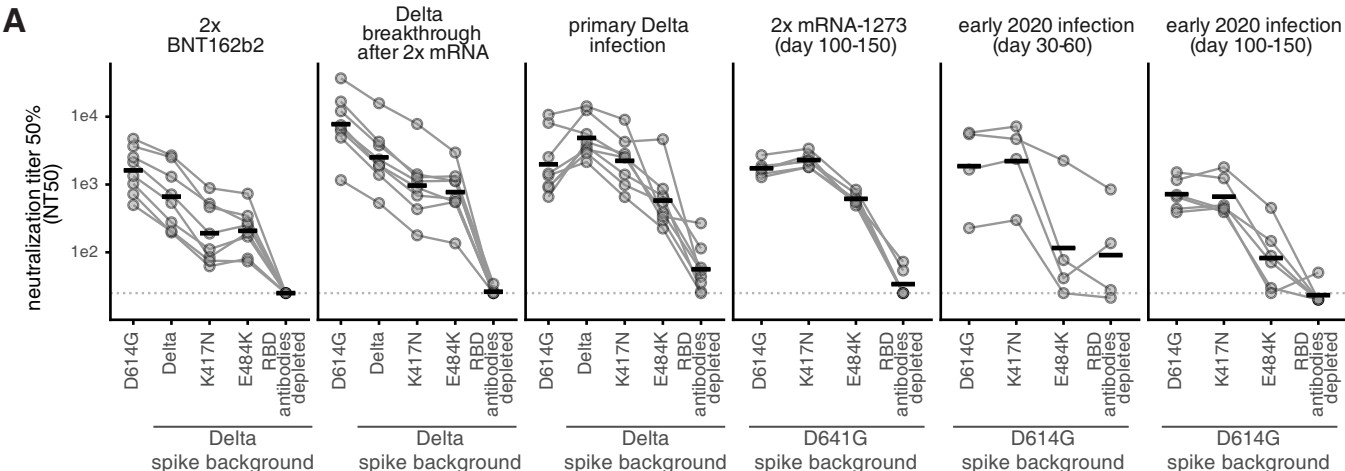

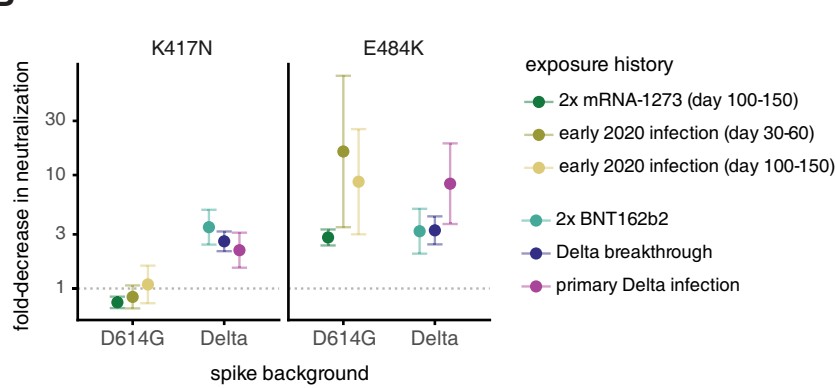

**Fig 6. Effect of mutations on neutralization by antibodies elicited by 2x BNT162b2 vaccination, Delta breakthrough after 2x mRNA vaccination, or primary Delta infection. (A)** Neutralization titer 50% (NT50) of spike-pseudotyped lentiviral particles by sera or plasmas from individuals with various exposure histories. The leftmost spike is D614G, which was circulating in 2020. The point mutations and depletion of RBD-binding antibodies are in the indicated spike background (D614G or Delta). Bars represent geometric mean. Dashed line indicates assay limit of detection. **(B)** Fold-decrease in neutralization by sera or plasmas from individuals with various exposure histories against K417N (left) or E484K (right) in the Delta and D614G spikes, compared to the homologous spike (dashed line at 1). Points represent the geometric mean across cohorts (NT50s plotted in (A)). Error bars represent geometric standard deviation. The neutralization assays with 2x BNT162b2, Delta breakthrough, and Delta primary infection were newly performed in this study, and assays with 2x mRNA-1273 and early 2020 plasmas were performed in [12,29] and are reanalyzed here. Full neutralization curves are in **S7 Fig** and numerical NT50s are in **S1 Data** and online at https://github.com/jbloomlab/SARS-CoV-2-RBD_Delta/blob/main/experimental_data/results/neut_titers/combined_neut_titers.csv.

and that Delta breakthrough infection boosts a Delta-specific neutralizing antibody response almost entirely focused on the RBD.

We mapped mutations that reduce binding of primary and breakthrough Delta infection-elicited polyclonal antibodies to the Delta RBD and found that antibody binding is most impacted by mutations to the class 1 and class 2 epitopes, including sites 417, 472, 478, and 484–486. In comparison, the antibody-binding response to infections with early 2020 viruses is affected by class 1, 2, and 3 mutations, whereas the Beta antibody response is more affected by class 2 and 3 mutations. Overall, Delta infection elicits a response that is more similar to early 2020 infection than Beta infection. This is consistent with other reports that Beta and Delta antibody responses are more dissimilar to each other than either is to the early 2020 antibody response [15–17,46].

We also compared the antibody response elicited by primary Delta infection to that elicited by a Delta breakthrough infection. The mutations that reduce binding of plasma antibodies from primary and breakthrough Delta infections to the Delta RBD are largely similar (e.g., those to class 1 and 2 epitopes). There are some subtle differences between primary and breakthrough Delta infection-elicited responses, however, with the Delta breakthrough responses sharing characteristics of both primary early 2020 and primary Delta infection responses. This is consistent with exposure to both spike variants among the individuals with a Delta breakthrough infection.

We also examined how the K417N and E484K mutations (in the class 1 and 2 epitopes, respectively) impact neutralization in the Delta spike background. K417N alone has little effect on neutralization in the D614G spike, but reduces neutralization by ~2–3-fold in the Delta background. This is likely because the class 1 epitope is subdominant to the E484-containing class 2 epitope in the neutralizing antibody response [23,24]. But, the effects of K417N can be unmasked when present in a spike background that already contains mutations that disrupt binding of some class 2 antibodies, such as E484K or L452R [1,15,47].

Our study has several limitations. First, each cohort in this study consists of a relatively small sample size. Due to patterns of SARS-CoV-2 viral circulation and competition among variants (e.g., early 2020, Beta, and Delta viruses circulated during different time periods and in different geographic regions), as well as the introduction of SARS-CoV-2 vaccination in 2021, the cohorts differ with regards to geographic location, age, and time of sample collection. Second, the antibody-escape mapping experiments were performed in a yeast-displayed deep mutational scanning system which only measures antibody binding to the RBD. There is not always a direct relationship between antibody binding and neutralization, as some epitopes are more immunodominant with regards to neutralization than others, and the relative contributions to neutralization of antibodies targeting different spike epitopes may depend on cell type and ACE2 expression level used in the experimental assays [3,34,35].

Despite these caveats, our results have important implications for future viral evolution and vaccine design. Due to the rapid antigenic evolution of SARS-CoV-2, public health experts must decide if and when to update SARS-CoV-2 vaccines [51,52]. But because each variant elicits an antibody response with its own susceptibilities to erosion by mutation, the impacts of mutations in future variants may depend on prior exposure history. This knowledge will help to understand which vulnerabilities may exist in a population to aid development of durable, broadly protective vaccines.

## Methods

### Ethics statement

All samples were obtained after written informed consent. Samples were collected from individuals enrolled in a prospective cohort study at the Africa Health Research Institute approved

by the Biomedical Research Ethics Committee at the University of KwaZulu–Natal (reference BREC/00001275/2020), the prospective longitudinal Hospitalized or Ambulatory Adults with Respiratory Viral Infections (HAARVI) cohort study in Washington State, USA approved by the University of Washington Institutional Review Board (protocol #STUDY00000959), or from the Benaroya Research Institute (BRI) Immune Mediated Disease Registry and Repository approved by the BRI Institutional Review Board (study number IRB08108).

## Key reagents

The SARS-CoV-2 RBD mutant libraries and unmutated parental plasmid are available upon request with completion of an MTA. The plasmid map for the unmutated Delta RBD in the yeast-display expression vector is at https://github.com/jbloomlab/SARS-CoV-2-RBD_Delta/blob/main/data/plasmids/3159_pETcon-SARS-CoV-2-RBD-L452R-T478K.gb. The plasmid encoding the SARS-CoV-2 spike gene used to generate pseudotyped lentiviral particles, HDM_Spikedelta21_D614G, is available from Addgene (#158762) and BEI Resources (NR-53765). The HDM_Spikedelta21_sinobiological_B.1.617.2 plasmid is available upon request, and the plasmid map is at https://github.com/jbloomlab/SARS-CoV-2-RBD_Delta/blob/main/data/plasmids/3181_HDM_Spikedelta21_sinobiological_B.1.617.2.gb. Further information and requests for reagents and resources should be directed to and will be fulfilled by Jesse Bloom (jbloom@fredhutch.org) upon completion of a materials transfer agreement.

## Description of cohort and ethics statement

Several cohorts were examined in this study. Cohort characteristics are in **S1 Table**. Results obtained for cohorts of individuals convalescent of primary or Delta breakthrough infection after 2x mRNA vaccination, or who received 2x BNT162b2 vaccination (uninfected) are new to this study. All results for all other cohorts are from previously published studies and are simply reanalyzed here [12,23,24,29]. In all studies, written informed consent was obtained from each participant. No participants report HIV infection or were HIV-positive if a test was performed. This was necessary due to the lentiviral basis of neutralization assays used in these studies.

Blood samples were obtained after written informed consent from adults with PCR-confirmed SARS-CoV-2 infection who were enrolled in a prospective cohort study at the Africa Health Research Institute approved by the Biomedical Research Ethics Committee at the University of KwaZulu–Natal (reference BREC/00001275/2020). Blood was sampled approximately 30 days post-symptom onset from 8 individuals infected with SARS-CoV-2 during the "third wave" of infections in South Africa from August through September 2021, when the Delta variant was detected in >90% of sequenced infections in the country [18]. Delta infection was corroborated by the experimental findings in this paper that all plasmas bound to Delta spike and RBD and better neutralized Delta spike-pseudotyped lentiviral particles relative to D614G particles. All participant samples had detectable antibody binding and neutralizing titers against the Delta SARS-CoV-2 spike.

Serum samples from individuals with a sequence-confirmed Delta breakthrough infection after two-dose mRNA vaccination were collected as a part of the prospective longitudinal Hospitalized or Ambulatory Adults with Respiratory Viral Infections (HAARVI) cohort study in Washington State, USA from July–September, 2021. All participants had a positive SARS-CoV-2 qPCR from a swab of the upper respiratory tract. 7 of the 8 participants had symptomatic infections (participant 273C was asymptomatic). The details of sequence confirmation of Delta infection are in [33]. Samples were collected approximately 30 days post-symptom onset (**S1 Table**). This work was approved by the University of Washington Institutional Review Board (protocol #STUDY00000959).

Serum samples from individuals vaccinated 2x with BNT162b2 were collected in Washington State, USA in January 2021. Samples were obtained from the Benaroya Research Institute (BRI) Immune Mediated Disease Registry and Repository, and were collected approximately 30 days post-dose 1 of vaccination (**S1 Table**). This study was approved by the BRI IRB, study number IRB08108.

Results from four previously published studies of individuals infected with early 2020 or Beta variant SARS-CoV-2 viruses, or vaccinated against SARS-CoV-2 are reanalyzed here (**S1 Table**) [12,23,24,29].

## Plasma separation from whole blood

Plasma was separated from EDTA-anticoagulated blood by centrifugation at 500 rcf for 10 min and stored at −80˚C. Aliquots of plasma samples were heat-inactivated at 56˚C for 60 min and clarified by centrifugation at 10,000 rcf for 5 min, after which the clear middle layer was used for experiments. Inactivated plasma was stored in single-use aliquots to prevent freeze–thaw cycles.

## Construction of Delta RBD yeast-displayed DMS library

Duplicate single-mutant site-saturation variant libraries were designed in the background of the spike receptor binding domain (RBD) from SARS-CoV-2 Delta variant (identical to that from Wuhan-Hu-1, Genbank accession number MN908947, residues N331-T531, with the addition of the L452R and T478K amino-acid substitutions), and produced by Twist Bioscience. The Genbank map of the plasmid encoding the unmutated SARS-CoV-2 Delta RBD in the yeast-display vector is available at https://github.com/jbloomlab/SARS-CoV-2-RBD_Delta/blob/main/data/plasmids/3159_pETcon-SARS-CoV-2-RBD-L452-T478K.gb. The site-saturation variant libraries were delivered as double-stranded DNA fragments by Twist Bioscience. The final unmutated DNA sequence delivered is:

```
tctgcaggctagtggtggaggaggctctggtggaggcggCCgcggaggcgga
gggtcggctagccatatgAATATCACGAACCTTTGTCCTTTCGGTGAGGTCTTCAATGCT
ACTAGATTCGCATCCGTGTATGCATGGAATAGAAAGAGAATTAGTAATTGTGTAGCGGACT
ACTCTGTACTTTATAACTCCGCCTCCTTCTCCACATTCAAGTGTTACGGTGTATCTCCCAC
CAAGTTGAATGATCTATGCTTTACAAACGTTTACGCCGATAGTTTCGTAATTAGAGGCGATG
AAGTGCGTCAGATCGCACCAGGCCAGACGGGCAAGATAGCAGACTATAATTATAAGCTG
CCTGATGACTTCACCGGCTGTGTGATAGCTTGGAACTCAAATAATCTAGATTCCAAGGTGGG
AGGCAATTACAATTATAGATACCGTCTGTTCCGTAAAAGCAATTTGAAACCATTTGAAAGAG
ACATTAGCACTGAAATTTATCAAGCAGGGTCCAAACCGTGCAACGGCGTAGAAGGCTTT
AACTGTTATTTCCCATTACAGTCTTATGGTTTCCAACCTACGAACGGAGTCGGGTATCAG
CCGTACAGGGTTGTGGTTCTTTCATTTGAACTGCTGCACGCGCCCGCAACCGTATGCGGG
CCGAAGAAATCAACGctcgaggggggcggttccgaacaaaagcttatttctgaagagga
cttgtaatagagatctgataacaacagtgtagatgtaacaaaatcgactttgttccca
ctgtactttagctcgtacaaaatacaatatacttttcatttctccgtaaacaaca
tgttttcccatgtaatatcctttttctatttttcgttccgttaccaacttttacacatacttta
tatagctattcacttctatacactaaaaaactaagacaattttaattttgctgcctgccata
tttcaatttgttataaattcctataatttatcctattagtagctaaaaaaagatgaa
tgtgaatcgaatcctaagagaatt
```

This sequence has 5' and 3' flanking sequences that are unmutated in the variant libraries (lower case). The uppercase portion is the RBD coding sequence, amino acids N331–T531 (Wuhan-Hu-1 spike numbering). The libraries were designed to contain all 20 amino acids at each site in the RBD. The design included stop codons at every other position for the first 81 positions, with the goal of 1% of variants in the library to contain premature stop codons,

which is useful for quality-control measures. The design targeted no more than one amino-acid mutation per variant. The variant gene fragments were PCR-amplified with these primers: 5'-tctgcaggctagtggtggag-3' and 5'-agatcggaagagcgtcgtgtagggaaagagtgtagatctcggtggtcgccgtatcat-taattctcttaggattcgattcacattc-3'. (primer-binding regions underlined in the sequence above). A second round of PCR was performed using the same forward primer (5'-tctgcaggctagtggtggag-3') and the reverse primer 5'-ccagtgaattgtaatacgactcactatagggcgaattg-gagctcgcggccgcnnnnnnnnnnnnnnnnnnagatcggaagagcgtcgtgtag-3' to append the Nx16 barcodes and add the overlapping sequences to clone into the recipient vector backbone as described in [53,54].

The barcoded variant gene fragments were cloned in bulk into the NotI/SacI-digested unmutated wildtype plasmid, as described in [12,53,54]. The Genbank plasmid map for the fully assembled, barcoded Delta RBD libraries (with the unmutated Delta RBD sequence) is available at https://github.com/jbloomlab/SARS-CoV-2-RBD_Delta/blob/main/data/plasmids/pETcon-SARS-CoV-2-RBD-Delta_lib-assembled.gb. The pooled, barcoded mutant libraries were electroporated into *E. coli* (NEB 10-beta electrocompetent cells, New England BioLabs C3020K) and plated at a target bottleneck of 100,000 variants per duplicate library, corresponding to >25 barcodes per mutant within each library. Colonies from bottlenecked transformation plates were scraped and plasmid purified. Plasmid libraries (10 μg plasmid per replicate library) were transformed into the AWY101 yeast strain [55] according to the protocol of Gietz and Schiestl [56].

## PacBio sequencing to link variant mutations and barcodes

As described by Starr et al. [53], PacBio sequencing was used to generate long sequence reads spanning the Nx16 barcode and RBD coding sequence. The 1079-basepair PacBio sequencing amplicons were prepared from the DNA plasmid libraries via NotI digestion, gel purification, and Ampure XP bead clean-up. Sample-specific barcoded SMRTbells were ligated using the SMRTbell Express Template Kit 2.0. The multiplexed libraries were sequenced on a PacBio Sequel IIe with a 20-hour movie collection time. Demultiplexed PacBio HiFi circular consensus sequences (CCSs) were generated on-instrument (Sequel IIe control software version 10.1.0.119549) and demultiplexed with lima using SMRTLink version 10.1.0.119588. HiFi reads are CCSs with $> = 3$ full passes and a mean quality score $Q > = 20$.

HiFi reads were processed using alignparse (version 0.2.6) [57] to determine each variant's mutations and the associated Nx16 barcode sequence, requiring no more than 45 nucleotide mutations from the intended target sequence, an expected 16-nt length barcode sequence, and no more than 4 mismatches across the sequenced portions of the vector backbone. Attribution of barcodes to library variants determined that the libraries contained all 3,819 possible single amino-acid mutations to the Delta RBD. Approximately 20% of barcodes in the duplicate libraries corresponded to the unmutated (wildtype) Delta RBD (**S3A Fig**). The libraries were designed to contain only wildtype and 1-amino acid mutations, but in some cases, multiple mutations per variant were stochastically introduced during the library synthesis process. These multi-mutant variants were excluded from downstream analysis of the effects of mutations on ACE2 binding, RBD expression, and plasma antibody binding.

## Determining the effects of mutations on RBD expression and ACE2 binding to filter the library for functional variants

The effects of each mutation on RBD expression on the surface of yeast and on ACE2 binding were measured essentially as described previously for the Wuhan-Hu-1 RBD [53]. Specifically, each biological replicate library was grown overnight at 30˚C in 45 mL synthetic defined

medium with casamino acids (SD-CAA: 6.7 g/L Yeast Nitrogen Base, 5.0 g/L Casamino acids, 2.13 g/L MES acid, and 2% w/v dextrose) at an initial OD600 of 0.4. To induce RBD surface expression, yeast were then diluted in SG-CAA+0.1%D (identical to SD-CAA except the dextrose is replaced with 2% w/v galactose supplemented with 0.1% dextrose) induction media at 0.67 OD600 and incubated at room temperature for 16–18 hours with mild agitation. For RBD expression experiments, 45 OD units of yeast were labeled in 1:100 diluted chicken-anti-Myc-FITC antibody (Immunology Consultants CMYC45F) to detect the RBD's C-terminal Myc tag. For ACE2-binding experiments, 12 OD units of yeast were incubated overnight at room temperature with monomeric biotinylated ACE2 (ACROBiosystems AC2-H82E8) across a concentration range of $10^{-13}$ M to $10^{-6}$ M at 1-log intervals (plus a 0 M ACE2 condition). Labeling volumes were increased at low ACE2 concentration to limit ligand depletion effects. Cells were then labeled with 1:100 diluted Myc-FITC to detect RBD expression and 1:200 Streptavidin-PE (Invitrogen S866) to detect binding of biotinylated ACE2.

Cells were processed on a BD FACSAria II and sorted into four bins from low to high RBD expression (measured by myc-FITC staining) or ACE2 binding (measured by streptavidin-PE fluorescence). The RBD expression sort bins were set such that bin 1 would capture 99% of unstained cells, and the remaining 3 bins divide the remainder of each mutant RBD library into equal tertiles. For ACE2 binding, bin 1 captured 95% of cells expressing unmutated RBD incubated with no ACE2 (0 M), and bin 4 captured 95% of cells expressing unmutated RBD incubated with a saturating amount of ACE2 ($10^{-6}$ M). Bins 2 and 3 equally divided the distance between the bin 1 upper and bin 4 lower fluorescence boundaries on a log scale. The frequency of each variant in each bin was determined by Illumina sequencing of RBD variant barcodes.

The effects of each mutation on RBD expression and ACE2 binding were determined as described in [53]. RBD mutant expression and ACE2 binding scores were calculated according to the equations in [53]. For ACE2 binding, a score of –1.0 corresponds to a 10-fold loss in affinity ($K_d$) compared to the wildtype RBD. For RBD expression, a score of –1.0 corresponds to a 10-fold reduction in mean RBD-myc-FITC fluorescence intensity. These measurements were used in downstream steps to computationally filter library variants that were highly deleterious for RBD expression or ACE2 binding and would likely represent spurious antibody-escape mutations (see below for details). The ACE2 binding and RBD expression scores for the single amino-acid mutations in the Delta RBD are available at https://github.com/jbloomlab/SARS-CoV-2-RBD_Delta/blob/main/data/final_variant_scores.csv.

As previously described, prior to performing the antibody-escape experiments, the yeast libraries were pre-sorted for RBD expression and binding to dimeric ACE2 (ACROBiosystems AC2-H82E6) to eliminate RBD variants that are completely misfolded or non-functional, such as those lacking modest ACE2 binding affinity [22]. Specifically, unmutated Delta RBD and each RBD mutant library were incubated with dimeric ACE2 at $10^{-8}$ M (a saturating concentration of ACE2 for unmutated Delta RBD). A FACS selection gate was set to capture 96% of cells expressing unmutated Delta RBD that were incubated with $10^{-10}$ M ACE2, to purge the mutant libraries of highly deleterious mutations (i.e., those that have <1% the affinity of unmutated Delta RBD). These pre-sorted yeast libraries containing RBD variants with at least nominal expression and ACE2 binding were used in downstream antibody-escape experiments (see below).

## Depleting plasma of nonspecific yeast-binding antibodies prior to antibody-escape experiments

Prior to the yeast-display deep mutational scanning, plasma samples were twice-depleted of nonspecific yeast-binding antibodies. AWY101 yeast containing an empty vector pETcon

plasmid were grown overnight, shaking, at 30˚C in SG-CAA media. Then, 100 μL of plasma samples were incubated, rotating, with 50 OD units of the yeast for 2 hours at room temperature in a total volume of 1mL. The yeast cells were pelleted by centrifugation, and the supernatant was transferred to an additional 50 OD units of yeast cells, and the incubation was repeated overnight at 4˚C. Before beginning the plasma-escape mapping experiments, the negative control yeast were pelleted by centrifugation and the supernatant (containing serum antibodies depleted of yeast-binding antibodies) was used in plasma-escape mapping.

## FACS sorting of yeast libraries to select RBD mutants with reduced binding by polyclonal antibodies

Plasma mapping experiments were performed in biological duplicate using the independent mutant RBD libraries, similarly to as previously described for monoclonal antibodies [22] and polyclonal plasma samples [23]. Mutant yeast libraries induced to express RBD were washed and incubated with diluted plasma for 1 hour at room temperature with gentle agitation. For each plasma, we chose a sub-saturating dilution such that the amount of fluorescent signal due to plasma antibody binding to RBD was approximately equal across samples. The exact dilution used for each plasma is given in **S4 Fig**. After the plasma incubations, the libraries were secondarily labeled for 1 hour with 1:100 fluorescein isothiocyanate-conjugated anti-MYC antibody (Immunology Consultants Lab, CYMC-45F) to label for RBD expression and 1:200 Alexa Fluor-647-conjugated goat anti-human-IgA+IgG+IgM (Jackson ImmunoResearch 109-605-064) to label for bound plasma antibodies. A flow cytometric selection gate was drawn to capture cells with the reduced plasma binding for their degree of RBD expression (**S3 Fig**). For each sample, approximately 10 million RBD$^+$ cells were processed on the BD FACSAria II cell sorter, with between 7 x 10$^5$ and 1.2 x 10$^6$ plasma-escaped cells collected per sample. Antibody-escaped cells were grown overnight in SD-CAA with 2% w/v dextrose + 100 U/mL penicillin + 100 μg/mL streptomycin to expand cells prior to plasmid extraction.

## DNA extraction and Illumina sequencing

Plasmid samples were prepared from 30 optical density (OD) units (1.6e8 colony forming units (cfus)) of pre-selection yeast populations and approximately 5 OD units (~3.2e7 cfus) of overnight cultures of plasma-escaped cells (Zymoprep Yeast Plasmid Miniprep II) as previously described [22]. The 16-nucleotide barcode sequences identifying each RBD variant were amplified by polymerase chain reaction (PCR) and prepared for Illumina sequencing as described in [53]. Specifically, a primer with the sequence 5′-AATGATACGGCGACCACCGAGA-3′ was used to anneal to the Illumina P5 adaptor sequence, and the PerkinElmer NextFlex DNA Barcode adaptor primers with the sequence 5′-CAAGCAGAAGACGGCATACGAGATxxxxxxxxGTGACTGGAGTT CAGACGTGTGCTCTTCCGATCT-3′ (where xxxxxxxx indicates the sample NextFlex index sequence) were used to anneal to the Illumina P7 adaptor sequence and append sample indexes for sample multiplexing. Barcodes were sequenced on an Illumina HiSeq 2500 or NextSeq 2000 with 50 bp single-end reads. To minimize noise from inadequate sequencing coverage, we ensured that each antibody-escape sample had at least 2.5x as many post-filtering sequencing counts as FACS-selected cells, and reference populations had at least 2.5e7 post-filtering sequencing counts.

## Analysis of deep sequencing data to compute each mutation's escape fraction

Escape fractions were computed as described in [22], with minor modifications as noted below. We used the dms_variants package (https://jbloomlab.github.io/dms_variants/, version

0.8.10) to process Illumina sequences into counts of each barcoded RBD variant in each pre-selection and antibody-escape population. For each plasma selection, we computed the escape fraction for each barcoded variant using the deep sequencing counts for each variant in the original and plasma-escape populations and the total fraction of the library that escaped antibody binding via the formula provided in [22]. Specifically:

$E_v = F \times (n_v^{post}/N_{post}) \div (n_v^{pre}/N_{pre})$ where $F$ is the total fraction of the library that escapes antibody binding (these fractions are given as percentages in S4B Fig), $n_v^{post}$ and $n_v^{pre}$ are the counts of variant $v$ in the RBD library after and before enriching for antibody-escape variants with a pseudocount of 0.5 added to all counts, and $N_{post} = \sum_v n_v^{post}$ and $N_{pre} = \sum_v n_v^{pre}$ are the total counts of all variants after and before the antibody-escape enrichment. These escape fractions represent the estimated fraction of cells expressing that specific variant that falls in the escape bin, such that a value of 0 means the variant is always bound by plasma and a value of 1 means that it always escapes plasma binding.

We then applied a computational filter to remove variants with >1 amino-acid mutation, low sequencing counts, or highly deleterious mutations that might cause antibody escape simply by leading to poor expression of properly folded RBD on the yeast cell surface [22,53]. Specifically, we removed variants that had ACE2 binding scores < −1.86 or expression scores < −0.75, after calculating mutation-level deep mutational scanning scores for this library as in [53]. An ACE2 binding score threshold of –1.86 retained 100% and an RBD expression score threshold of –1.0 retained 97.3% of all RBD mutations observed > = 50x in GISAID as of Aug. 1, 2021 (S3D Fig).

We also removed all mutations where the wildtype residue was a cysteine. There were 2,251 out of the possible 3,666 mutations to non-disulfide bond residues in the RBD that passed these computational filters.

The reported antibody-escape scores throughout the paper are the average across the libraries; these scores are also in S3 Data. Correlations in final single-mutant escape scores are shown in S5 Fig.

For plotting and analyses that required identifying RBD sites of strong escape, we considered a site to mediate strong escape if the total escape (sum of mutation-level escape fractions) for that site exceeded the median across sites by >10-fold, and was at least 10% of the maximum for any site. Full documentation of the computational analysis is at https://github.com/jbloomlab/SARS-CoV-2-RBD_Delta.

## Differences between composition and analysis of Delta RBD libraries and Wuhan-Hu-1 libraries

Importantly, because the Delta libraries were generated using a different method than the Wuhan-Hu-1 RBD libraries, which is fully described in [53], the analysis of deep sequencing data to compute each mutation's escape fraction is also different. The newly generated Delta libraries (like the Beta RBD mutant libraries described in [12]) were ordered from Twist Bioscience to have one amino-acid mutation per variant, whereas the Wuhan-Hu-1 libraries were generated in-house with a PCR-based approach, with an average of 2.7 mutations per variant [53]. Because there were often multiple mutations per variant for the Wuhan-Hu-1 libraries, global epistasis modeling was used to deconvolve the effects of single amino-acid mutations on antibody binding [22,23], whereas for the Delta libraries, the measurements for single-mutant variants were used directly (occasional variants with multiple mutations were discarded) to calculate antibody escape. This is the same analysis framework that was used for the Beta RBD libraries described in [12].

## Generation of pseudotyped lentiviral particles

HEK-293T (American Type Culture Collection, CRL-3216) cells were used to generate SARS-CoV-2 spike-pseudotyped lentiviral particles and 293T cells stably overexpressing ACE2 and TMPRSS2 (kind gift of Carol Weiss, FDA) were used to titer the SARS-CoV-2 spike-pseudotyped lentiviral particles and to perform neutralization assays (see below).

For experiments involving D614G spike, we used spike-pseudotyped lentiviral particles that were generated essentially as described in [58], using a codon-optimized SARS-CoV-2 spike from Wuhan-Hu-1 strain that contains a 21-amino-acid deletion at the end of the cytoplasmic tail [59] and the D614G mutation that is now predominant in human SARS-CoV-2 [25]. The plasmid encoding this spike, HDM_Spikedelta21_D614G, is available from Addgene (#158762) and BEI Resources (NR-53765), and the full sequence is at (https://www.addgene.org/158762). For experiments involving Delta spike, we cloned a human codon-optimized Delta spike gene into the HDM expression plasmid. The spike amino acid-sequence matches that of the Wuhan-Hu-1 spike with the following substitutions: T19R, G142D, del156-157, R158G, L452R, T478K, D614G, P681R, D950N. This plasmid map is available online at https://github.com/jbloomlab/SARS-CoV-2-RBD_Delta/blob/main/data/plasmids/3181_HDM_Spikedelta21_sinobiological_B.1.617.2.gb.

To generate spike-pseudotyped lentiviral particles [58], $6 \times 10^5$ HEK-293T (ATCC CRL-3216) cells per well were seeded in 6-well plates in 2 mL D10 growth media (Dulbecco's Modified Eagle Medium with 10% heat-inactivated fetal bovine serum, 2 mM l-glutamine, 100 U/mL penicillin, and 100 μg/mL streptomycin). 24 hours later, cells were transfected using BioT transfection reagent (Bioland Scientific) with a Luciferase_IRES_ZsGreen backbone, Gag/Pol lentiviral helper plasmid (BEI Resources NR-52517), and wild-type or mutant SARS-CoV-2 spike plasmids. Media was changed to fresh D10 at 24 hours post-transfection. At ~60 hours post-transfection, viral supernatants were collected, filtered through a 0.45 μm surfactant-free cellulose acetate low protein-binding filter, and stored at −80˚C.

## Titering of pseudotyped lentiviral particles

Titers of spike-pseudotyped lentiviral particles were determined as described in [58] with the following modifications. 100 μL of serially diluted spike-pseudotyped lentiviral particles were added to $1.25 \times 10^4$ 293T-ACE2 cells (BEI Resources NR-52511), grown overnight in 50 μL of D10 growth media in a 96-well black-walled poly-L-lysine coated plate (Greiner Bio-One, 655936). Relative luciferase units (RLU) were measured 50 hours post-infection (Promega Bright-Glo, E2620) in the infection plates with a black back-sticker (Thermo Fisher Scientific, NC9425162) added to minimize background.

## Neutralization assays

293T-ACE2 cells (BEI Resources NR-52511) were seeded at $1.25 \times 10^4$ cells per well in 50 μL D10 in poly-L-lysine coated, black-walled, 96-well plates (Greiner 655930). 24 hours later, pseudotyped lentivirus supernatants were diluted to ~$2 \times 10^5$ RLU per well (determined by titering as described above) and incubated with a range of dilutions of plasma for 1 hour at 37˚C. 100 μL of the virus-antibody mixture was then added to cells. At 50 hours post-infection, luciferase activity was measured using the Bright-Glo Luciferase Assay System (Promega, E2610). Fraction infectivity of each plasma antibody-containing well was calculated relative to a no-plasma well inoculated with the same initial viral supernatant in the same row of the plate. We used the neutcurve package (https://jbloomlab.github.io/neutcurve version 0.5.7) to calculate the inhibitory concentration 50% ($IC_{50}$) and the neutralization titer 50% ($NT_{50}$),

which is $1/IC_{50}$, of each plasma against each virus by fitting a Hill curve with the bottom fixed at 0 and the top fixed at 1.

## Depletion of RBD-binding antibodies from polyclonal sera

Magnetic beads conjugated to the SARS-CoV-2 Wuhan-Hu-1 RBD (ACROBiosystems, MBS-K002) or Delta RBD (ACROBiosystems, MBS-K037) were prepared according to the manufacturer's protocol. Beads were resuspended in ultrapure water at 1 mg beads/mL and washed 3 times in phosphate-buffered saline (PBS) with 0.05% w/v bovine serum albumin (BSA). Beads were then resuspended in PBS with 0.05% BSA at 1 mg beads per mL. Beads were incubated with human plasma at a 3:1 ratio of beads:plasma, rotating overnight at 4°C or for 2 hours at room temperature. A magnet (MagnaRack Magnetic Separation Rack, Thermo Fisher Scientific, CS15000) was used to separate antibodies that bind RBD from the supernatant, and the supernatant was collected. A mock depletion was performed by adding an equivalent volume of PBS + 0.05% BSA and rotating overnight at 4°C or for 2 hours at room temperature. Three or four rounds of depletions were performed to ensure full depletion of RBD-binding antibodies. Note that these assays were performed in 293T cells over-expressing human ACE2, which emphasize contributions of ACE2-competitive antibodies to viral neutralization [3,34,60].

## Measurement of plasma binding to RBD or spike by enzyme-linked immunosorbent assay (ELISA)

The IgG ELISAs for spike protein and RBD were conducted as previously described [61]. Briefly, ELISA plates were coated with recombinant Wuhan-Hu-1 (purified and prepared as described in [61]) or Delta RBD (ACRO Biosystems SPD-C52Hh) and RBD (ACROBiosystems, SPD-C52Hp) antigens at 0.5 μg/mL. Five 3-fold serial dilutions of sera beginning at 1:100 or 1:500 were performed in PBS with 0.1% Tween with 1% nonfat dry milk. Secondary labeling was performed with goat anti-human IgG-Fc horseradish peroxidase (HRP) (1:3000, Bethyl Labs, A80-104P). Antibody binding was detected with TMB/E HRP substrate (Millipore Sigma, ES001) and 1 N HCl was used to stop the reaction. $OD_{450}$ was read on a Tecan infinite M1000Pro plate reader.

## Data visualization

The static logo plot visualizations of the escape maps in the paper figures were created using the dmslogo package (https://jbloomlab.github.io/dmslogo, version 0.6.2) and in all cases the height of each letter indicates the escape fraction for that amino-acid mutation calculated as described above. For each sample, the y-axis is scaled to be the greatest of (a) the maximum site-wise escape metric observed for that sample, (b) 20x the median site-wise escape fraction observed across all sites for that plasma, or (c) an absolute value of 1.0 (to appropriately scale samples that are not noisy but for which no mutation has a strong effect on antibody binding). For each set of logo plots (Figs 3B and 5A and S6), sites of "strong escape" for any of the included plasmas or sera are highlighted in pink on the x-axis of the line plots, and featured in the corresponding logo plots at right. We define these sites of "strong escape" as those where the total escape (sum of mutation-level escape fractions) for that site exceeded the median across sites by > 10-fold, and was at least 10% of the maximum for any site. Sites 417, 452, 477, 478, 484, and 501 have been added to logo plots due to their polymorphism among circulating viruses. The code that generates these logo plot visualizations is available at https://github.com/jbloomlab/SARS-CoV-2-RBD_Delta/blob/main/results/summary/escape_profiles.md. In many of the visualizations, the RBD sites are categorized by epitope region [37] and colored

accordingly. We define the class 1 epitope as residues 403+405+406+417+420+421+453+455–460+473–478+486+487+489+503+504, the class 2 epitope as residues 472+483–485+490–495, the class 3 epitope to be residues 341+345+346+352–357+396+437–452+466+468+496+498–501, and the class 4 epitope as residues 365–372+378+382–386+408.

For the static structural visualizations in the paper figures, the RBD surface (PDB 6M0J) was colored by the site-wise escape metric at each site, with white indicating no escape and red scaled to be the same maximum used to scale the y-axis in the logo plot escape maps, determined as described above. We created interactive structure-based visualizations of the escape maps using dms-view [62] that are available at https://jbloomlab.github.io/SARS-CoV-2-RBD_Delta/. The logo plots in these escape maps can be colored according to the deep mutational scanning measurements of how mutations affect ACE2 binding or RBD expression as described above.

The multidimensional scaling in Figs 4 and 5 that projects the antibody-escape maps into two dimensions was performed using the Python scikit-learn package, version 1.0.1. We first computed the similarity between the escape maps of each pair of plasmas or antibodies: Let $x_{a1}$ be the vector of the total site escape values at each site for antibody $a1$. The similarity in escape between antibodies $a1$ and $a2$ is the dot product of the total site escape vectors after normalizing each vector to have a Euclidean norm of one; namely, the similarity is $\left(\frac{x_{a1}}{||x_{a1}||}\right) \cdot \left(\frac{x_{a2}}{||x_{a2}||}\right)$. Thus, the similarity is one if the total site escape is identical for the two antibodies, and zero if the escape is at completely distinct sites. Then, we computed the dissimilarity as one minus the similarity, then performed metric multidimensional scaling with two components on the dissimilarity matrix exactly as defined in [22].

## Supporting information

**S1 Table. Summary characteristics of cohorts examined in this study.**
(DOCX)

**S1 Fig. Enzyme-linked immunosorbent assay (ELISA) of plasmas or sera before and after depletion of Wuhan-Hu-1 or Delta RBD-binding antibodies.** Plasma or serum binding to the (**A**) Delta RBD or (**B**) Delta spike after mock or depletion of Delta RBD-binding antibodies. (**C**) Plasma or serum binding to D614G RBD after mock or depletion of D614G RBD-binding antibodies. Note that the D614G RBD is identical to the Wuhan-Hu-1 RBD.
(EPS)

**S2 Fig. Plasma neutralization of spike-pseudotyped lentiviral particles before and after depletion of Wuhan-Hu-1 (D614G) or Delta RBD-binding antibodies.** Plasma neutralization titer 50% (NT50) of D614G or Delta spike-pseudotyped lentiviral particles for mock depletion or depletion of Wuhan-Hu-1 (D614G) or Delta RBD-binding antibodies. The labels above the plots indicate which RBD-binding antibodies were depleted and which spikes were pseudotyped on lentiviral particles. The same data are plotted by individual serum/plasma sample in (**A**) or grouped by cohort in (**B**). A subset of these results are shown grouped by cohort in **Fig 2**. For plasmas depleted of Delta RBD-binding antibodies tested against D614G spike, neutralizing activity could be due to RBD-binding antibodies that do not bind the Delta RBD, or due to non-RBD-binding antibodies. The full neutralization curves are in **S7A Fig**. All neutralization data are in **S1 Data** and online at https://github.com/jbloomlab/SARS-CoV-2-RBD_Delta/blob/main/experimental_data/results/neut_titers/combined_neut_titers.csv.
(EPS)

**S3 Fig. Generation of the Delta RBD mutant libraries and measurements of effects of mutations on ACE2 binding and RBD expression. (A)** Schematic showing the Delta RBD

mutant library design. A site-saturation variant library was generated in the Delta RBD background, targeting one amino-acid mutation per variant. Nx16 unique DNA barcodes were appended to the variant gene fragments. The Nx16 barcodes were linked to their associated RBD mutations by PacBio circular consensus sequencing (CCS). The plasmid library DNA was transformed into yeast cells. In downstream experiments, the Nx16 barcodes are sequenced by short-read Illumina sequencing. The tables at right indicate key library statistics. **(B)** FACS gating scheme used to measure effects of RBD mutations on expression on the yeast cell surface (left) or on binding to monomeric ACE2 (right). The ACE2 binding and expression scores are in **S2 Data** and online at https://github.com/jbloomlab/SARS-CoV-2-RBD_Delta/blob/main/results/final_variant_scores/final_variant_scores.csv. These results are summarized in greater detail in [36]. **(C)** Correlations between biological independent replicate library measurements of the effects of single mutations on ACE2 binding and RBD expression, measured as described in [53]. See **Methods** for experimental details. **(D)** Thresholds on the ACE2 binding and RBD expression scores (dashed orange lines) for the Delta mutant library to computationally filter highly deleterious variants that may represent spurious antibody-escape mutations. We aimed to retain most mutations that have been observed $> = 50$ times in sequenced SARS-CoV-2 isolates. The x-axis categorizes mutations by their number of observations in GISAID [65] as of Aug. 1, 2021. An ACE2 binding score threshold of $> = -$1.86 ($10^{-1.86}$ binding compared to unmutated Delta RBD, which is a 72.4-fold loss in binding affinity) and an RBD expression score of $> = -0.75$ ($10^{-0.75}$ expression compared to unmutated Delta RBD, which is a 5.6-fold loss in RBD expression) were chosen, which filter comparable numbers of mutations as in prior Wuhan-Hu-1 experiments [23,43]. These filters retain 100 and 97.3% of mutations, respectively, that have been observed $> = 50$ times in sequenced SARS-CoV-2 isolates. **(E)** Relationship between the ACE2 binding and RBD expression scores for the Delta RBD library compared to those previously published for the Wuhan-Hu-1 library in Starr, et al. (2020) [53]. The computational filters used for antibody-escape experiments for the Wuhan-Hu-1 [23] and Delta libraries are dashed orange lines. Each dot is one mutation, and mutations to disulfide bonds are shown in red. A key difference is that for the previously published Wuhan-Hu-1 experiments, dimeric rather than monomeric ACE2 was used [53]. (EPS)

**S4 Fig. Deep mutational scanning approach to map mutations that reduce binding of polyclonal plasma antibodies to the Delta RBD. (A)** Schematic of the approach. Libraries of yeast expressing Delta RBD mutants (measured via a C-terminal MYC tag, green star) were incubated with polyclonal antibodies from plasmas or sera and fluorescence-activated cell sorting (FACS) was used to enrich for cells expressing RBD with reduced antibody binding, as detected using an IgA+IgG+IgM secondary antibody. Deep sequencing was used to quantify the frequency of each mutation in the pre-selection and antibody-escape cell populations. The escape fraction represents the fraction of cells expressing RBD with that mutation that fell in the antibody escape FACS bin. Experimental and computational filters were used to remove RBD mutants that were misfolded or unable to bind the ACE2 receptor. **(B)** Top left: Representative plots of nested FACS gating strategy used for all plasma selection experiments to select for RBD-expressing single cells. Samples were gated by SSC-A versus FSC-A, SSC-W versus SSC-H, and FSC-W versus FSC-H) that also express RBD (FITC-A vs. FSC-A). Bottom left: The RBD mutant libraries were sorted to retain cells expressing variants that bound to ACE2 with at least nominal affinity. The RBD mutant libraries were incubated with dimeric ACE2 at $10^{-8}$ M and FACS-enriched for cells that bound ACE2 with at least as much fluorescence intensity as cells expressing unmutated Delta RBD that were incubated with $10^{-10}$ M ACE2, to purge the mutant libraries of highly deleterious mutations (i.e., those that have <1%

the affinity of unmutated Delta RBD). This retained ~93% of RBD-expressing fractions of the libraries. Right: FACS gating strategy for one of two independent libraries to select cells expressing RBD mutants with reduced binding by polyclonal sera (cells in blue), as measured with an anti-IgA+IgG+IgM secondary antibody. Gates were set manually during sorting. Selection gates were set to capture cells with reduced antibody binding for their degree of RBD expression. FACS scatter plots were qualitatively similar between the two libraries. Serum dilutions used in selections are indicated. SSC-A, side scatter-area; FSC-A, forward scatter-area; SSC-W, side scatter-width; SSC-H, side scatter-height; FSC-W, forward scatter-width; FSC-H, forward scatter height; FITC-A, fluorescein isothiocyanate-area.
(EPS)

**S5 Fig. Correlations of antibody-escape scores between replicates. (A)** Site- and **(B)** mutation-level correlations of escape scores between two independent biological replicate libraries. The complete antibody-escape scores are in **S3 Data** and online at https://github.com/jbloomlab/SARS-CoV-2-RBD_Delta/blob/main/results/supp_data/aggregate_raw_data.csv.
(EPS)

**S6 Fig. Complete maps of mutations that reduce binding of polyclonal serum antibodies to the Delta RBD.** Delta mutant library escape maps for sera from individuals with **(A)** primary or breakthrough Delta infections after 2x mRNA vaccination (replicated here from **Figs 3 and 5** to facilitate direct comparison) or **(B)** 2x BNT162b2 vaccination. Sites of strong antibody escape (see Methods) for any of the 8 plasmas in A or B are highlighted with pink in the line plots at left and shown in the logo plots at right. Sites 417, 452, 484, 477, 478, and 501 are included in the logo plots whether or not they are sites of strong escape due to their high frequency in circulating viral isolates. **(C)** The average site-total antibody escape for plasmas from individuals vaccinated with 2x mRNA-1273 against the Wuhan-Hu-1 mutant libraries (previously published in [29]) or 2x BNT162b2 against the Delta mutant libraries. Key epitopes are shaded. **(D)** The average site-total antibody escape mapped to the surface of the Wuhan-Hu-1 RBD (PDB 6M0J, [64]), with red indicating the site with the strongest antibody escape, and white indicating no escape. Key sites are labeled, with labels colored according to antibody epitope. The vaccine sera in B–D are from individuals who were not exposed to the Delta spike via infection or vaccination. The L452R mutation in the Delta RBD can disrupt antibody binding to both the class 2 and class 3 antibody epitopes [24,42], and thus the 2x BNT162b2 (x Delta mutant libraries) plasmas are mostly escaped by mutations in the class 4 epitope (including sites 365, 383, 384) or a non-canonical class 3 epitope that includes site 357. The antibody-escape maps against the Delta RBD mutant libraries are newly generated in this study, whereas the 2x mRNA-1273 antibody-escape maps against the Wuhan-Hu-1 RBD mutant libraries were first reported in [29] and are reanalyzed here.
(PDF)

**S7 Fig. Neutralization of plasmas against spike-pseudotyped lentiviral particles. (A)** Raw neutralization curves for assays shown in **Figs 2 and 6 and S2**. **(B)** Left: Table summarizing the effect of the K417N and E484K mutations in the Delta and D614G spike backgrounds on neutralization. Right: summary plot of the geometric mean across cohorts of the fold-decrease in neutralization compared to Delta spike (left axis) or D614G spike (right axis). Error bars represent geometric standard deviation. The neutralization assays with 2x BNT162b2, Delta breakthrough, and Delta primary infection were newly performed in this study, and assays with 2x mRNA-1273 and early 2020 plasmas were performed in [12,29] and are reanalyzed here. Neutralization titers are in **S1 Data** and online at https://github.com/jbloomlab/SARS-CoV-2-RBD_Delta/blob/main/experimental_data/results/neut_titers/combined_neut_titers.csv.
(EPS)

**S1 Data. Neutralization titers for all neutralization assays performed or analyzed in this study.** This includes assays newly performed in this study on sera or plasmas from individuals with 2x BNT162b2 vaccination, Delta breakthrough after 2x mRNA vaccination, or Delta primary infections before and after depletion of D614G (Wuhan-Hu-1) or Delta RBD-binding antibodies or against point mutants in the Delta spike background. This table also includes previously published neutralization titers from individuals infected with early 2020 viruses [23], vaccinated 2x with mRNA-1273 [29], and individuals with a primary Beta infection [12]. This table is also available at: https://github.com/jbloomlab/SARS-CoV-2-RBD_Delta/blob/main/experimental_data/results/neut_titers/combined_neut_titers.csv (CSV)

**S2 Data. The effects of all single amino-acid mutations in the Delta RBD on ACE2 binding and RBD expression.** These results are also published in [36]. This file is also available at: https://github.com/jbloomlab/SARS-CoV-2-RBD_Delta/blob/main/results/final_variant_scores/final_variant_scores.csv. (CSV)

**S3 Data. Plasma-escape scores for sera or antibodies mapped against the Wuhan-Hu-1, Beta, or Delta RBD single-mutant libraries.** The serum-escape scores for sera/plasmas mapped against the Delta RBD libraries are new to this study. The data for antibodies or sera/plasmas mapped against the Wuhan-Hu-1 or Beta RBD libraries were previously published [12,22–24,29,42–44]. This table is also available at https://github.com/jbloomlab/SARS-CoV-2-RBD_Delta/blob/main/results/supp_data/aggregate_raw_data.csv. (CSV)

## Acknowledgments

We thank Dolores Covarrubias, Andy Marty, the Genomics and Flow Cytometry core facilities at the Fred Hutchinson Cancer Research Center. We also thank all study participants for their generous participation and contribution to this work. We acknowledge the BRI COVID-19 Research Team for participant recruitment and retention efforts.

## Author Contributions

**Conceptualization:** Allison J. Greaney, Alex Sigal, Jesse D. Bloom.

**Formal analysis:** Allison J. Greaney.

**Funding acquisition:** Jesse D. Bloom.

**Investigation:** Allison J. Greaney, Rachel T. Eguia.

**Methodology:** Allison J. Greaney, Tyler N. Starr.

**Resources:** Khadija Khan, Nicholas Franko, Jennifer K. Logue, Sandra M. Lord, Cate Speake, Helen Y. Chu, Alex Sigal.

**Software:** Allison J. Greaney.

**Supervision:** Cate Speake, Helen Y. Chu, Alex Sigal, Jesse D. Bloom.

**Validation:** Allison J. Greaney, Rachel T. Eguia.

**Visualization:** Allison J. Greaney.

**Writing – original draft:** Allison J. Greaney, Jesse D. Bloom.

**Writing – review & editing:** Allison J. Greaney, Tyler N. Starr, Khadija Khan, Nicholas Franko, Jennifer K. Logue, Sandra M. Lord, Cate Speake, Helen Y. Chu, Alex Sigal, Jesse D. Bloom.

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
