## [Decision Letter · Decision Letter 0]

18 Apr 2022

Dear Dr Bloom,

Thank you very much for submitting your manuscript "The SARS-CoV-2 Delta variant induces an antibody response largely focused on class 1 and 2 antibody epitopes" for consideration at PLOS Pathogens. As with all papers reviewed by the journal, your manuscript was reviewed by members of the editorial board and by several independent reviewers. The reviewers appreciated the attention to an important topic. Based on the reviews, we are likely to accept this manuscript for publication, providing that you modify the manuscript according to the review recommendations.

Sincerely,

Meike Dittmann, Ph.D.

Associate Editor

PLOS Pathogens

Michael Diamond

Section Editor

PLOS Pathogens

Kasturi Haldar

Editor-in-Chief

PLOS Pathogens

orcid.org/0000-0001-5065-158X

Michael Malim

Editor-in-Chief

PLOS Pathogens

orcid.org/0000-0002-7699-2064

Reviewer Comments (if any, and for reference):

Reviewer's Responses to Questions

**Part I - Summary**

Reviewer #1: The manuscript by Greaney et al. explores the variation of antibody responses following different types of infection. This is the latest in a series of manuscripts by the Bloom lab that uses their clever deep mutational scan approach to characterize antibody responses that the types of mutations that can evade them. The major conclusion is that Delta infections predominantly lead to type 1 and type 2 antibody classes. This is an excellent manuscript.

Reviewer #2: Greaney and colleagues investigate the impact of differently SARS-CoV-2 exposure history on serum antibody specificity. This work directly compares serum from early pandemic infections, vaccinations, likely Beta infections, likely Delta infections, and delta breakthrough infections. They use deep mutational scanning to identify sites that disrupt serum antibody binding and find that exposure history influences the spectrum of “escape” mutations. The work is important, timely, expertly performed and well-written. I cannot offer meaningful improvements. Well done to all authors.

Reviewer #3: This paper by Greaney et al builds on this team's well-established deep mutational scanning approach to show that infection by Delta triggers antibodies that differ subtly from those elicited by previous variants. Its elegant work and carefully described, though I note I am not able to evaluate the computational aspects of this study.

**Part II – Major Issues: Key Experiments Required for Acceptance**

Reviewer #1: The one major comment is that as a reader I felt like there was a blaring omission. In the beginning and the end of the manuscript they show data from both infected, vaccinated, and vaccinated breakthroughs. However, when they characterize the Delta responses in Figure 5, they only look at Delta primary infections and vaccinated with delta breakthroughs, but they don't show the data for vaccinated without a delta infection. It felt like a 2-legged stool. It's hard to believe they don't have this data. I think the authors really need to provide this data or at least provide the reader an explanation for why it is not included.

Reviewer #2: (No Response)

Reviewer #3: None

**Part III – Minor Issues: Editorial and Data Presentation Modifications**

Reviewer #1: Minor comment. I could not find where the authors explicitly said how they decided what amino acid changes to include in their mutational maps (Fig 4B and 5A). I assume these were the most pronounced effects, but it would help to say so explicitly.

Reviewer #2: (No Response)

Reviewer #3: The title could be more informative in capturing variant specific nuances

Figure 2 – it seems that though all responses are focussed on RBD, the slope of the curve varies in depletion experiments. Specifically Beta seems to have flatter curves. Is this meaningful ?

Can the authors rule out prior infection? If not this should be mentioned in limitations

PLOS authors have the option to publish the peer review history of their article (what does this mean?). If published, this will include your full peer review and any attached files.

Reviewer #1: No

Reviewer #2: No

Reviewer #3: No

Figure Files:

Data Requirements:

Reproducibility:

References:

---

## [Editor Report · Decision Letter 1]

15 May 2022

Dear Dr Bloom,

We are pleased to inform you that your manuscript 'The SARS-CoV-2 Delta variant induces an antibody response largely focused on class 1 and 2 antibody epitopes' has been provisionally accepted for publication in PLOS Pathogens.

Best regards,

Meike Dittmann, Ph.D.

Associate Editor

PLOS Pathogens

Michael Diamond

Section Editor

PLOS Pathogens

Kasturi Haldar

Editor-in-Chief

PLOS Pathogens

orcid.org/0000-0001-5065-158X

Michael Malim

Editor-in-Chief

PLOS Pathogens

orcid.org/0000-0002-7699-2064
---

## [Editor Report · Acceptance letter]

10 Jun 2022

Dear Dr Bloom,

We are delighted to inform you that your manuscript, "The SARS-CoV-2 Delta variant induces an antibody response largely focused on class 1 and 2 antibody epitopes," has been formally accepted for publication in PLOS Pathogens.

Best regards,

Kasturi Haldar

Editor-in-Chief

PLOS Pathogens

orcid.org/0000-0001-5065-158X

Michael Malim

Editor-in-Chief

PLOS Pathogens

orcid.org/0000-0002-7699-2064